# Effects of Slit Lamp Examination on Tear Osmolarity in Normal Controls and Dry Eye Patients

**DOI:** 10.3390/bioengineering12101124

**Published:** 2025-10-20

**Authors:** Myung-Sun Song, Jooye Park, Hae Jung Paik, Dong Hyun Kim

**Affiliations:** 1Department of Ophthalmology, Korea University College of Medicine, Seoul 02841, Republic of Korea; 5098love@naver.com; 2Department of Ophthalmology, Gil Medical Center, Gachon University College of Medicine, Incheon 21999, Republic of Korea; evilanne@naver.com (J.P.); hjpaik@gilhostpial.com (H.J.P.)

**Keywords:** dry eye disease, tear osmolarity, slit lamp examination

## Abstract

**Background/Objective**: Tear hyperosmolarity is the main triggering factor in the immunopathogenesis of dry eye disease (DED). Tear osmolarity is known as the relevant metric to evaluate DED severity; however, measuring tear osmolarity after slit lamp examination (SLE) is known as a contraindication due to variability. In this study, we investigated the effects of SLE and fluorescein staining (FS) on the variabilities of tear osmolarity. **Methods:** In this prospective observational study sixty-five patients were enrolled in the study, comprising 31 healthy controls and 34 DED patients. The tear osmolarity was measured in the right eye using the TearLab^®^ system. The initial measurements were performed to establish baseline values before SLE, and additional measurements were performed after 20 s of SLE and followed by 20 s of SLE+FS. There were five-minute intervals between measurements. A correlation analysis was performed between OSDI score, tear film break-up time (TBUT), and tear osmolarity. A linear mixed-effects model was also applied to account for repeated measures and inter-subject variability. **Results:** The mean ages of the control and DED group were 31.3 ± 11.5 and 50.5 ± 15.5 years. Increased tear osmolarity was significantly associated with greater OSDI score and lower TBUT only in DED patients, but not in normal controls (OSDI:R = 0.378/*p* = 0.030, TBUT:R = −0.543/*p* = 0.011). The mean tear osmolarities in the normal controls were 298.3 ± 11.3, 299.1 ± 13.3, and 297.0 ± 12.6 mOsm/L at baseline (group 1), after SLE (group 2), and after SLE+FS (group 3), respectively, with no significant difference (*p* = 0.379). However, there was a significant difference in the tear osmolarities of the three groups in the DED patients (296.1 ± 11.5, 296.5 ± 11.0, and 291.2 ± 11.3 mOsm/L for groups 1–3, respectively/*p* < 0.001). The tear osmolarity of group 3 was significantly lower than those of groups 1 and 2 in the DED patients (*p* = 0.010/0.016). After FS, the mean tear osmolarity decreased by 4.9 ± 9.2 mOsm compared to the baseline in DED group. **Conclusions:** Tear osmolarity was only decreased in DED patients after SLE+FS, whereas it was unaffected in normal control subjects. Increased tear osmolarity in only DED patients correlated with increased symptom scores and decreased TBUT. These fluctuations in tear osmolarity reflect compromised tear film homeostasis in DED, highlighting the need to contextualize osmolarity data with clinical DED parameters.

## 1. Introduction

Dry eye disease (DED) is a highly prevalent and multifactorial disorder of the ocular surface, affecting millions worldwide and significantly diminishing patients’ quality of life. The Tear Film and Ocular Surface Society (TFOS) Dry-Eye Workshop (DEWS) II defined DED as a “multifactorial disease of the ocular surface, characterized by a loss of homeostasis of the tear film accompanied by ocular symptoms, in which tear film instability and hyperosmolarity, ocular surface inflammation and damage, and neurosensory abnormalities play etiological roles” [1]. The diagnosis of DED is based on subjective ocular symptoms, quantified with a questionnaire, such as the Dry Eye Questionnaire and ocular surface disease index (OSDI), and objective signs, such as decreased tear film break-up time (TBUT), positive ocular surface staining (OSS), and reduced tear volume [2]. Despite its high prevalence, the diagnosis and assessment of DED remain challenging due to the inconsistent relationship between clinical signs and patient-reported symptoms.

Tear hyperosmolarity is one of the main triggering factors in the immunopathogenesis of DED [3]. The TFOS DEWS II diagnostic methodology report showed that a positive result of tear osmolarity is considered to be ≥308 mOsm/L or an interocular difference > 8 mOsm/L [4]. Meanwhile, Asia Dry Eye Society (ADES) emphasize the unstable tear film and the diagnostic value of the ophthalmologists’ observations [5]. Tear osmolarity is not a main diagnostic component in the ADES report. Nonetheless, the TearLab^®^ (TearLab™ Corp., San Diego, CA, USA), a device for measuring tear osmolarity, has been widely used in DED patient care. The TearLab^®^ consists of a collection pen for tear collection, a reader where the collection pen is docked, and single-use disposable test cards, and a temperature-compensated impedance is used to measure the tear osmolarity in the collected tear fluid sample [6]. To date, the TearLab^®^(TearLab Corp, San Diego, CA, USA) is the gold standard for measuring tear osmolarity with a cutoff value of 308 mOsm/L required to diagnose DED [4]. Several studies have established the diagnostic value of tear osmolarity due to its correlation with DED severity; an increase in tear osmolarity reflects an increase in DED severity [7,8,9,10]. Kim et al. showed that tear osmolarity measurements using the TearLab^®^ system reflected the signs and symptoms of ocular Sjögren’s syndrome [7]. Tear osmolarity also showed good performance, especially in severe dry eye, and increased stepwise according to DED severity [8]. However, Bunya et al. demonstrated that the clinical utility of tear osmolarity measurement in dry eye patients is limited by either high intra-session variability or poor correlation with subjective symptoms [9,10].

According to the TearLab^®^ manual, collecting tears within 15 min after slit lamp examination (SLE) or fluorescein staining (FS) is contraindicated [6]. Accordingly, tear osmolarity tests should be performed before performing an ophthalmologic examination. The manufacturer guidelines for the TearLab^®^ system, which specifies avoiding slit lamp manipulation or dye application prior to baseline measurements to prevent reflex tearing or fluorescein-induced osmolarity elevation (reported to increase readings by 8–12 mOsm/L) [6]. In clinical practice, the timing and sequence of diagnostic tests are also critical, as procedures such as slit lamp examination or fluorescein staining can transiently alter tear osmolarity measurements, potentially leading to diagnostic inaccuracies. However, considering the busy clinical environment such as South Korea, patients may be dissatisfied with receiving the tear osmolarity test before seeing the doctor for evaluating DED. Despite these challenges, tear osmolarity remains an objective and reproducible tool for evaluating DED severity, and its integration into a multimodal diagnostic approach may enhance the accuracy and reliability of DED diagnosis and management. Therefore, further research is warranted to clarify the optimal use of tear osmolarity testing in the context of routine ophthalmologic examinations and to better understand the factors influencing its variability in both healthy individuals and patients with DED. No previous studies have investigated the changes in tear osmolarity after the SLE or FS. We hypothesize that slit lamp examination and fluorescein staining will not induce a statistically significant or clinically meaningful increase in tear osmolarity measurements, and the change in osmolarity and its variability following these procedures will not differ significantly between normal controls and patients with dry eye disease. Therefore, this study investigated the effects of SLE or FS on tear osmolarity and the variability in tear osmolarity after SLE or FS was also compared between normal controls and patients with DED.

## 2. Materials and Methods

This prospective, single-center, observational cohort study was conducted at the Ocular Surface Disease Specialty Clinic of Gachon University Gil Medical Center (GUGMC), a tertiary care institution recognized for its advanced diagnostic and therapeutic protocols in anterior segment disorders. The study spanned from September 2019 to July 2020, to ensure adequate enrollment of participants and minimize seasonal confounding factors. Ethical oversight was rigorously maintained through approval by the Institutional Review Board (IRB) of Gachon University Gil Medical Center (IRB No. GAIRB2019-068). Written informed consent was obtained from all participants after comprehensive explanations of study objectives, procedures, and potential risks. The research protocol aligned with the ethical principles outlined in the 8th revision of the Declaration of Helsinki, prioritizing patient autonomy, confidentiality (de-identified data handling), and minimization of physical/psychological harm. Enrollment criteria were systematically applied through electronic health record (EHR) screening, and all clinical assessments utilized standardized equipment calibrated under GUGMC’s quality assurance program.

Tear osmolarity was measured using the TearLab^®^ Osmolarity System (TearLab™ Corp., San Diego, CA, USA), following the manufacturer’s standardized protocol [6]. DED was diagnosed according to the new diagnostic guideline of Korean Dry Eye Society, emphasizing subjective symptoms and tear film instability, with an Ocular Surface Disease Index (OSDI) score of 13 or higher and TBUT of 7 s or less as key criteria [11]. Patients with an active ocular infection, eyelid deformity, corneal opacity, and history of contact lenses or ocular surgery within six months were excluded from the study. The additional exclusion criteria were an ongoing pregnancy, systemic immunosuppressive drug use, primary or secondary Sjogren’s syndrome, corneal refractive surgery within three years, and punctal plug placement within three months. DED patients had been prescribed 0.05% cyclosporine (RestasisTM, Allergan, Inc., CA, USA) twice daily, and 0.15% hyaluronic acid (New Hyaluni ophthalmic solution 0.15%, Taejoon Pharm., South Korea) had been used as needed. All measurements were performed in a controlled clinical environment during morning hours (8:00–10:00 a.m.) to minimize diurnal variation, and patients were instructed to refrain from topical medication use, ocular manipulation, and excessive screen exposure for at least two hours prior to testing. Calibration of the TearLab^®^ device was verified weekly using certified control solutions. DED parameters, such as the TBUT and OSS, were measured by an experienced corneal specialist (DH Kim). The ocular staining score (OSS) was determined using the Oxford scheme scale (0–5 points) [12]. OSDI, OSS, and TBUT values were used to distinguish between DED and normal control groups. Subjects with an OSDI score of 13 or more and a TBUT of 7 s or less were enrolled with a dry eye group.

Tear osmolarity was measured sequentially in the right eye using the TearLab^®^ Osmolarity System under strictly controlled environmental conditions (ambient temperature: 20–25 °C, humidity: 40–50%) to minimize variability. Baseline tear sample collection was obtained prior to any ophthalmic procedures. Subsequently, tear samples were collected immediately after slit lamp examination (SLE) and again after SLE with fluorescein staining (SLE+FS), with a standardized 5 min interval between each phase to allow tear film stabilization. (Figure 1) The Haag Streit BQ 900 slit lamp (Haag-Streit AG, Bern, Switzerland) was utilized at maximal illumination intensity with a 30° angle between the microscope and illumination system, ensuring optimal visualization of the ocular surface. The slit width and length were maximum (14 mm). SLE included a 20 s evaluation of TBUT and ocular surface staining (OSS), with fluorescein applied via a sterile strip moistened with saline to the inferior temporal bulbar conjunctiva. Patients were instructed to remove eyeglasses during examinations, and natural blinking was permitted. Room lighting remained consistent, and all procedures were conducted in a single session to reduce inter-test variability. The purpose of this protocol was to quantify the impact of routine clinical interventions on tear osmolarity dynamics, particularly relevant given the device’s sensitivity to environmental variables such as ambient conditions and measurement timing. The examiner performing the tear osmolarity measurements was masked to the participant’s group status (normal control versus DED patient) to minimize potential observation bias.

All statistical analyses were conducted using SPSS^®^ Statistics version 18.0 (IBM Corp., Armonk, NY, USA). Continuous variables were summarized as mean ± standard deviation (SD), and categorical data as frequencies or percentages. To assess changes in tear osmolarity across the three measurement phases (baseline, post-SLE, post-SLE+FS), a linear mixed-effects model (LMM) was used to account for repeated measures and inter-subject variability. Post hoc pairwise comparisons between timepoints were adjusted using the Bonferroni method to control for multiple testing. Correlations between changes in tear osmolarity and clinical parameters such as TBUT, OSDI, and OSS were evaluated using the Pearson correlation coefficient (r), with scatterplots generated for visualization and 95% confidence intervals (CI) reported. Assumptions of normality and homoscedasticity were checked, and sensitivity analyses using Spearman’s rank correlation were performed as needed. Statistical significance was defined as a two-tailed *p*-value < 0.05.

## 3. Results

A total of 65 participants were enrolled in this prospective study, comprising 31 healthy controls and 34 patients diagnosed with DED. The baseline characteristics of both groups are detailed in Table 1. The mean age of the control group was 31.3 ± 11.5 years (range: 19–58), while the DED group was significantly older, with a mean age of 50.5 ± 15.5 years (range: 28–76), reflecting a statistically significant difference (*p* < 0.001). The DED patients demonstrated a markedly higher mean OSDI score of 41.4 ± 16.9 points, compared to 8.1 ± 3.5 points in the control group (*p* < 0.001). TBUT was also significantly reduced in the DED patients (3.8 ± 1.6 s) relative to controls (6.5 ± 2.8 s, *p* < 0.001). In contrast, ocular surface staining (OSS), as assessed by the Oxford scale, did not differ significantly between the groups (DED: 0.3 ± 0.6; control: 0.1 ± 0.4; *p* = 0.211), nor did the proportion of participants with a history of prior corneal refractive surgery (DED: 8.8%; control: 22.6%; *p* = 0.124).

Figure 2 illustrates the correlation between baseline tear osmolarity, OSDI scores, and TBUT across study cohorts. In the normal control group, Pearson correlation analysis revealed no statistically significant associations between tear osmolarity and OSDI scores (R = 0.144, *p* = 0.438) or TBUT (R = 0.202, *p* = 0.274), suggesting that physiological tear film homeostasis in healthy eyes minimizes osmolarity-symptom/stability linkages. Conversely, the DED group exhibited distinct patterns: a positive correlation between tear osmolarity and OSDI scores (R = 0.378, *p* = 0.030), indicating that higher osmolarity aligns with worse patient-reported symptoms, and a strong negative correlation between osmolarity and TBUT (R = −0.543, *p* = 0.011), reflecting reduced tear film stability as hyperosmolarity escalates.

Table 2 shows the differences in the tear osmolarities of the control and DED groups at baseline (group 1), after SLE (group 2), and after FS and SLE (SLE+FS) (group 3). The mean tear osmolarities of the control group were 298.3 ± 11.3, 299.1 ± 13.3, and 297.0 ± 12.6 (mOsm/L) at baseline, after SLE, and after SLE+FS, respectively. In the normal controls, there was no significant difference in the tear osmolarities among the three groups (*p* = 0.379, repeated measure ANOVA). However, there was a significant difference in the tear osmolarities of the three groups in the DED patients (296.1 ± 11.5, 296.5 ± 11.0, and 291.2 ± 11.3 mOsm/L at baseline, after SLE, and after SLE+FS; *p* < 0.001, repeated measure ANOVA). When comparing differences in tear osmolarity between groups 1–3 in the DED group, notable differences in tear osmolarity were observed between group 1 and 3 and group 2 and 3 were −4.9 ± 9.2 and −5.2 ± 10.4 mOsm/L, respectively.

Figure 3 demonstrates the serial changes in tear osmolarity among the three groups between normal controls and DED patients. In normal controls (Figure 3A), tear osmolarity remained remarkably stable across all three measurement conditions, with forest plots confirming no statistically significant differences between any measurement pairs. In contrast, DED patients (Figure 3B) exhibited significant changes in tear osmolarity across the three measurement conditions (*p* < 0.001, repeated measure ANOVA), especially with a notable decrease following fluorescein staining. While no significant difference was observed between baseline and After SLE in DED patients, tear osmolarity significantly decreased after SLE+FS compared to both the baseline (*p* = 0.010) and after SLE (*p* = 0.016) measurements, as indicated by asterisks on the bar graph and confirmed by the forest plots showing positive mean differences with confidence intervals not crossing zero.

The normal control and DED groups in this study exhibited a notable age difference, which may influence tear osmolarity variability. However, in the DED group, a subgroup analysis restricted to participants aged 20–30 years demonstrated tear osmolarity values of 304.8 mOsm/L at baseline, 302.3 mOsm/L after SLE, and 284.6 mOsm/L after SLE+FS (Friedman test, *p* < 0.0001). Similar to the overall results in DED group, a decrease in tear osmolarity was observed only after SLE+FS, a pattern that was distinct from that of the normal control group.

## 4. Discussion

This prospective study demonstrated a differential impact of routine ophthalmic procedures on tear osmolarity measurements between normal controls and dry eye disease (DED) patients. Slit lamp examination (SLE) alone did not significantly alter tear osmolarity in either group. However, SLE with fluorescein staining (SLE+FS) exclusively affected DED patients, causing a mean decrease of 4.9 ± 9.2 mOsm/L compared to baseline, while normal controls remained unaffected. Furthermore, higher tear osmolarity only in DED patients correlated significantly with increased OSDI scores and reduced TBUT, highlighting the pathophysiological vulnerability of the DED tear film to diagnostic procedures.

Measuring tear osmolarity is widely used in diagnosing DED because it is a quick and painless technique. The TFOS DEWS II pathophysiology subcommittee has reported that tear hyperosmolarity is mainly involved in initiating and perpetuating DED [4]. Several studies have reported that, based on the TearLab^®^, the mean tear osmolarity in normal adults was 298–304 mOsm/L [13,14,15]. In this study, the baseline tear osmolarities of the healthy controls and DED patients were not significantly different (normal controls: 298.3 ± 11.3, DED patients: 296.1 ± 11.5). This may be since only level 1 or 2 DED patients were included in the study or may be due to the relatively small sample size. Even so, for normal controls, SLE or FS had no effect on osmolarity, but for DED patients SLE+FS significantly impacted osmolarity results.

Our results demonstrated that tear osmolarity in DED patients exhibited a significant positive correlation with OSDI scores (*R* = 0.378, *p* = 0.030) and a strong negative correlation with TBUT (*R* = −0.543, *p* = 0.011), aligning with that tear hyperosmolarity correlated with DED severity. However, the relationship between tear osmolarity and individual clinical DED parameters remains inconsistent across studies. Versura et al. showed significant correlations between tear osmolarity and OSDI and TBUT, using Tearlab^®^ [15]. In a study of tear osmolarity using I-pen, Park et al. showed significant correlations between tear osmolarity and OSDI and TBUT as well [16]. These two studies included both normal and DED patients, like our study. Mathews et al.’s study including both normal and DED patients showed significant positive correlations between tear osmolarity and OSDI, but for TBUT the correlation did not show statistical significance [17]. Conversely, tear osmolarity was weakly correlated with conjunctival (*r* = 0.18; *p* < 0.001) and corneal staining scores (*r* = 0.17; *p* < 0.001), TBUT (*r* = 0.06; *p* = 0.03), and Schirmer test score (*r* = −0.07; *p* = 0.02) but not with OSDI scores (*r* = 0.03; *p* = 0.40) in DREAM study (2023) [18]. In a study of Sjogren-related DED patients, tear osmolarity had a significant negative correlation to TBUT, but its correlation to OSDI was not significant [19]. All together, these indicate that there is no single best DED index, and that the various DED parameters including symptoms and signs should be taken together to correlate with the severity of the disease.

This study revealed that SLE alone did not significantly alter tear osmolarity in either DED patients or normal controls. However, the combination of SLE with fluorescein staining (SLE+FS) induced a notable dilutive effect on tear osmolarity specifically in DED patients, while no such changes were observed in normal controls. This discrepancy may be attributed to the compromised integrity of the lacrimal functional unit (LFU) in DED patients, which comprises the lacrimal glands, ocular surface, eyelids, and associated sensory/motor nerves responsible for maintaining tear film homeostasis [20]. In normal controls, an intact LFU likely counteracts the dilutive effects of fluorescein-staining procedures through reflex tearing, relevant blinking, and a neurophysiological response to ocular surface irritation that rapidly secretes balanced tears to neutralize the stimulus. Conversely, in DED patients, LFU dysfunction characterized by reduced lacrimal gland output, diminished corneal sensitivity, and impaired neural feedback loops fail to elicit adequate reflex tearing and tear secretion. Consequently, the fluorescein application remains diluted in the tear film, transiently lowering the measured tear osmolarity. This mechanism aligns with prior studies demonstrating that DED patients exhibit blunted reflex tearing responses to mechanical or chemical stimuli [21,22,23]. Furthermore, chronic inflammation in DED may exacerbate LFU impairment, rendering the ocular surface less responsive to procedural perturbations. These findings underscore the critical role of LFU integrity in buffering against external stressors and highlight the pathophysiological vulnerability of DED patients to diagnostic interventions that transiently alter tear composition. In addition, the clinical significance of our finding is primarily the differential pattern of change between the groups, which reflects impaired tear film homeostasis in DED patients. Specifically, only DED patients experienced a notable drop (4.9 to 5.3 mOsm/L) after SLE+FS. However, when considering established diagnostic criteria, such as the TearLab^®^ interocular difference threshold of >8 mOsm/L, this 5 mOsm/L decrease is unlikely to be a major clinical confounder that would dramatically shift a DED diagnosis or significantly alter patient management.

This study has several limitations. First, the relatively small sample size (65 participants) may limit the statistical power to detect subtle changes in tear osmolarity and reduces the generalizability of findings to broader populations, particularly given the heterogeneity of DED. Second, while we controlled for major confounders, other clinically variables-including systemic comorbidities (e.g., autoimmune disorders, diabetes) and environmental factors (e.g., screen time, ambient humidity), potentially introducing bias. Third, tear osmolarity was measured only in the right eye; therefore, assessment of inter-eye variability—a recognized diagnostic parameter for DED (interocular difference > 8 mOsm/L)—was not feasible. Fourth, the significant age disparity between the DED group (mean 50.5 years) and controls (mean 31.3 years) introduces confounding, as aging is independently associated with DED, meibomian gland dysfunction, and increased ocular surface inflammation. Fifth, the single-center design may have introduced site-specific biases in patient recruitment and procedural standardization. Sixth, there is a significant age difference between normal controls and DED patients. Seventh, Keratograph 5M device were not used in this study. Eighth, the study did not account for diurnal variations in tear osmolarity, which are more pronounced in DED patients, nor did it assess the impact of prolonged eyelid manipulation during fluorescein staining on reflex tearing. Ninth, the impact of systemic medications/conditions on tear osmolarity should be considered. Despite these limitations, this study is the first prospective investigation to systematically evaluate the effects of routine ophthalmic examinations (SLE/SLE+FS) on tear osmolarity variability, providing critical insights into pre-analytical factors that influence osmolarity measurements. We believe that the tear osmolarity values after ophthalmic examinations can be appropriately applied in the clinical aspects through this study.

## 5. Conclusions

In conclusion, tear osmolarity was affected only in DED patients when slit lamp examination with fluorescein staining was applied, while slit lamp examination alone did not significantly alter tear osmolarity. Furthermore, there were no significant differences in the tear osmolarities before and after SLE or SLE+FS in normal controls. A decrease of approximately 5 mOsm/L, compared to the baseline, was observed after SLE+FS; however, SLE itself did not affect the variability of the tear osmolarity in DED patients. While these findings support the ideal workflow of measuring osmolarity before SLE+FS to prevent procedure-induced changes, our results also offer a pragmatic approach for busy clinics. Since the mean decrease is relatively small compared to established diagnostic thresholds (e.g., >8 mOsm/L interocular difference), clinicians can cautiously interpret post-SLE+FS readings, as the absolute value still provides meaningful data for tracking DED severity. Since several environmental factors influence tear osmolarity, further studies are needed to investigate the changes in tear osmolarity in relation to environmental changes. In addition, higher tear osmolarity in only DED patients, not in normal controls, significantly correlated with increased symptom scores and reduced TBUT. These differences in tear osmolarity changes highlight the homeostatic disruption of the tear film in DED patients, and clinical DED parameters should be considered in the clinical interpretation of tear osmolarity data.

## Figures and Tables

**Figure 1 bioengineering-12-01124-f001:**
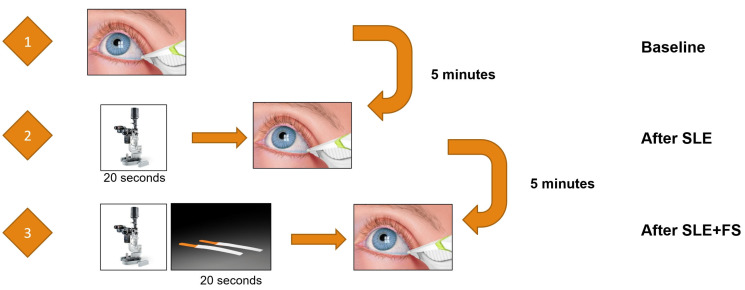
A schematic diagram illustrating the sequence for measuring tear osmolarity. SLE: slit lamp examination; FS; fluorescein stain.

**Figure 2 bioengineering-12-01124-f002:**
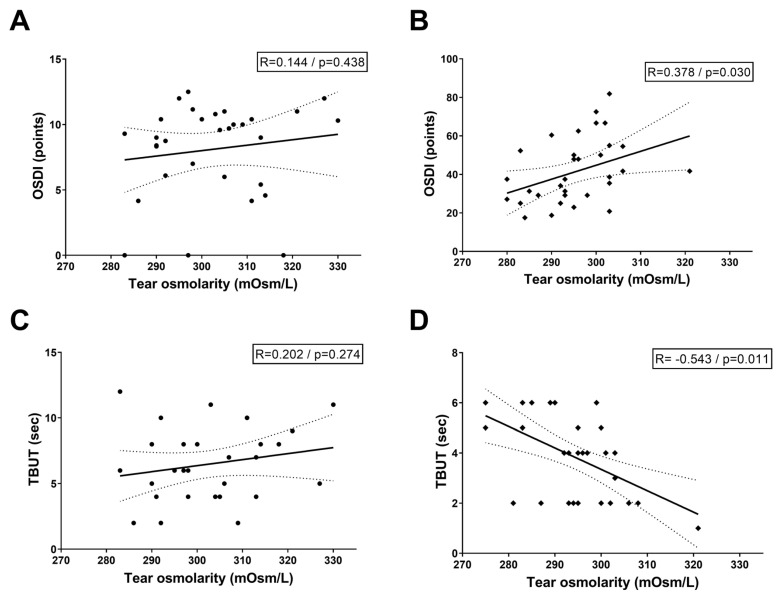
Correlation between ocular surface disease index (OSDI) score and average tear film osmolarity in normal control (**A**) and DED patients (**B**). Correlation between average tear film break-up time (TBUT) and average tear film osmolarity in normal control (**C**) and DED patients (**D**). DED: dry eye disease. Black lines: correlation slope, dotted line: range of standard deviation.

**Figure 3 bioengineering-12-01124-f003:**
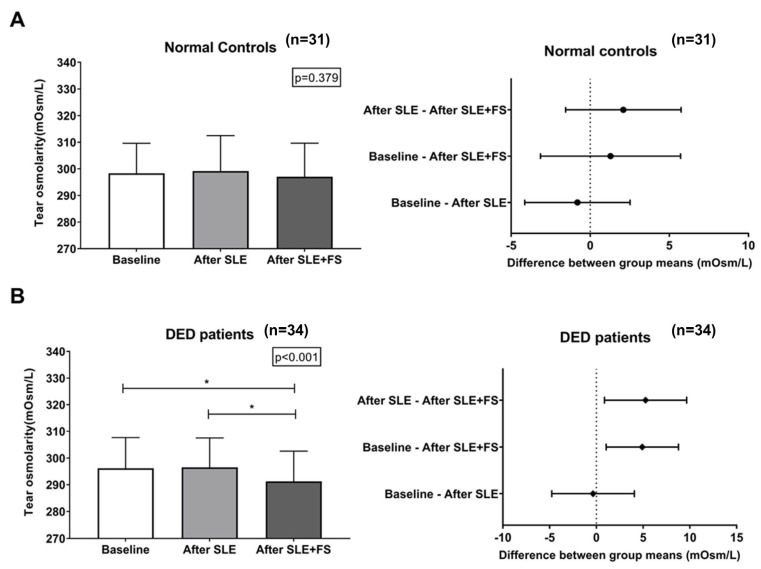
Comparison of tear osmolarity after slit lamp examination and fluorescein staining. (**A**) Normal controls; (**B**) DED patients. DED: dry eye disease. *: *p* < 0.05.

**Table 1 bioengineering-12-01124-t001:** Baseline characteristics of patients with normal control groups and DED patients.

	Normal Controls (*n* = 31)	DED Patients (*n* = 34)	*p* Value *
Age (years)	31.3±11.5	50.5 ± 15.5	<0.001
Sex (M/F)	8/23	7/27	0.618 †
OSDI (points)	8.1 ± 3.5	41.4 ± 16.9	<0.001
TBUT (seconds)	6.5 ± 2.8	3.8 ± 1.6	<0.001
Ocular staining score (points)	0.1 ± 0.4	0.3 ± 0.6	0.211
Corneal refractive surgery (+/−)	7/24	3/31	0.124 †

DED: dry eye disease; OSDI: ocular surface disease index; TBUT: tear film breakup time. *: independent *t*-test; †: chi square test. Values are presented as mean ± SD.

**Table 2 bioengineering-12-01124-t002:** Changes in tear osmolarity according to slit lamp examination and fluorescein staining.

	Group 1	Group 2	Group 3	*p* Value ^†^
Normal controls (mOsm/L) **(*n* = 31)**	298.3 ± 11.3	299.1 ± 13.3	297.0 ± 12.6	0.379
DED patients (mOsm/L) **(*n* = 34)**	296.1 ± 11.5	296.5 ± 11.0	291.2 ± 11.3	<0.001

Group 1: baseline; Group 2: after SLE; Group 3: after SLE+FS. SLE: slit lamp examination; FS; fluorescein stain; DED: dry eye disease. ^†^: repeated measure ANOVA. Values are presented as mean ± SD.

## Data Availability

The datasets used and/or analyzed during the current study are available from the corresponding author on reasonable request.

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
