# Peer review of "Effects of Slit Lamp Examination on Tear Osmolarity in Normal Controls and Dry Eye Patients"

_bioengineering, 2025, doi:10.3390/bioengineering12101124_

Round 1
Reviewer 1 Report
Comments and Suggestions for Authors
File attached

Author Response
Reviewer 1
Thank you for the opportunity to review your manuscript. This study addresses an important and
underexplored question—the effect of slit lamp examination and fluorescein staining on tear
osmolarity—using a prospective design. The work is clinically relevant and may influence
diagnostic workflows in the evaluation of dry eye disease (DED). Overall, the study is well-conducted
and well-presented. I have a few suggestions to improve clarity and strengthen the
paper:
- Thank you for your comment.
- Introduction
- Consider condensing the introduction by reducing repeated references to osmolarity and TearLab®.
- A clear, concise statement of your primary hypothesis at the end of the introduction would help frame the study.
- Thank you for your comment. We corrected the introduction as you suggested and reduced repeated sentences and references to osmolarity and TearLab®. In addition, we made concise primary hypothesis at the end of the introduction as below. (Introduction 2nd and 3rd paragraph)
“Tear hyperosmolarity is one of the main triggering factors in the immunopathogenesis of DED.[3] TFOS DEWS II diagnostic methodology report showed that a positive result of tear osmolarity is considered to be ≥308 mOsm/L or an interocular difference >8 mOsm/L.[4] Meanwhile, Asia Dry Eye Society (ADES) emphasize the unstable tear film and the diagnostic value of the ophthalmologists’ observations.[5] Tear osmolarity is not a main diagnostic component in ADES report. Nonetheless, the TearLab® (TearLab™ Corp., San Diego, CA, USA), a device for measuring tear osmolarity, has been widely used in DED patient care. The TearLab® consists of a collection pen for tear collection, a reader where the collection pen is docked, and single-use disposable test cards, and a temperature-compensated impedance is used to measure the tear osmolarity in the collected tear fluid sample.[6] Up to date, the TearLab® is the gold standard for measuring tear osmolarity with a cutoff value of 308 mOsm/L required to diagnose DED.[13] Several studies have established the diagnostic value of tear osmolarity due to its correlation with DED severity; an increase in tear osmolarity reflects an increase in DED severity.[7-10] Kim et al. showed that tear osmolarity measurements using the TearLab® system reflected the signs and symptoms of ocular Sjögren’s syndrome.[7] Tear osmolarity also showed good performance, especially in severe dry eye, and increased stepwise according to DED severity.[8] However, Bunya et al. demonstrated that the clinical utility of tear osmolarity measurement in dry eye patients is limited by either high intra-session variability or poor correlation with subjective symptoms.[9-10]
According to the TearLab® manual, collecting tears within 15 min after slit lamp examination (SLE) or fluorescein staining (FS) is contraindicated.[6] Accordingly, tear osmolarity tests should be performed before performing an ophthalmologic examination. The manufacturer guidelines for the TearLab® system, which specifies avoiding slit lamp manipulation or dye application prior to baseline measurements to prevent reflex tearing or fluorescein-induced osmolarity elevation (reported to increase readings by 8–12 mOsm/L). [6] In clinical practice, the timing and sequence of diagnostic tests are also critical, as procedures such as slit lamp examination or fluorescein staining can transiently alter tear osmolarity measurements, potentially leading to diagnostic inaccuracies. However, considering the busy clinical environment such as South Korea, patients may be dissatisfied with receiving the tear osmolarity test before seeing the doctor for evaluating DED. Despite these challenges, tear osmolarity remains an objective and reproducible tool for evaluating DED severity, and its integration into a multimodal diagnostic approach may enhance the accuracy and reliability of DED diagnosis and management. Therefore, further research is warranted to clarify the optimal use of tear osmolarity testing in the context of routine ophthalmologic examinations and to better understand the factors influencing its variability in both healthy individuals and patients with DED. No previous studies have investigated the changes in tear osmolarity after the SLE or FS. We hypothesize that slit lamp examination and fluorescein staining will not induce a statistically significant or clinically meaningful increase in tear osmolarity measurements, and the change in osmolarity and its variability following these procedures will not differ significantly due to the action of lacrimal functional unit (LFU). Therefore, this study investigated the effects of SLE or FS on tear osmolarity and the variability in tear osmolarity after SLE or FS was also compared between normal controls and patients with DED.”
- Methods
- Please provide details on sample size determination (e.g., power calculation) to reinforce methodological rigor.
- Thank you for your important comment. Unfortunately, this was a pilot study addressing the complete lack of prior data on tear osmolarity changes following slit lamp examination (SLE) and fluorescein staining (FS). Therefore, conventional power analysis and sample size calculation were infeasible. So we followed common empirical guidelines for pilot studies, where 10 to 15 or sometimes 30 to 50 participants are suggested, we selected a sample size sufficient in this novel study.
- The age mismatch between normal controls and DED patients may confound results. If feasible, include statistical adjustment for age or a more explicit discussion of its impact.
- Thank you for your important comment. We fully agree that the age mismatch between the normal controls and the DED patients is a potential confounding factor. We attempted to recruit age-matched normal controls but found this extremely challenging given the recruitment context of a tertiary care university hospital. In the subgroup analysis with adjustment for age, the trend was similar as below in the Result section despite of smaller sample size. In addition, we described the lack of strict age matching as a limitation of this study in the Discussion section.
“The normal control and DED groups in this study exhibited a notable age difference, which may influence tear osmolarity variability. However, in the DED group, a subgroup analysis restricted to participants aged 20–30 years demonstrated tear osmolarity values of 304.8 mOsm/L at baseline, 302.3 mOsm/L after SLE, and 284.6 mOsm/L after SLE + FS (Friedman test, P < 0.0001). Similar to the overall results in DED group, a decrease in tear osmolarity was observed only after SLE + FS, a pattern that was distinct from that of the normal control group.
“
“Fourth, the significant age disparity between the DED group (mean 50.5 years) and controls (mean 31.3 years) introduces confounding, as aging is independently associated with DED, meibomian gland dysfunction, and increased ocular surface inflammation.”
- Clarify whether examiners were masked to participant group status. Masking would help reduce bias.
- Thank you for your important comment. We masked the examiner from the participants. These were described in the methods section as below (Methods 3rd paragraph)
“Tear osmolarity was measured sequentially in the right eye using the TearLab® Osmolarity System under strictly controlled environmental conditions (ambient temperature: 20–25°C, humidity: 40–50%) to minimize variability. Baseline tear sample collection was obtained prior to any ophthalmic procedures. Subsequently, tear samples were collected immediately after slit lamp examination (SLE) and again after SLE with fluorescein staining (SLE+FS), with a standardized 5-minute interval between each phase to allow tear film stabilization. The Haag Streit BQ 900 slit lamp (Haag-Streit AG, Bern, Switzerland) was utilized at maximal illumination intensity with a 30° angle between the microscope and illumination system, ensuring optimal visualization of the ocular surface. SLE included a 20-second evaluation of TBUT and ocular surface staining (OSS), with fluorescein applied via a sterile strip moistened with saline to the inferior temporal bulbar conjunctiva. Patients were instructed to remove eyeglasses during examinations, and natural blinking was permitted. Room lighting remained consistent, and all procedures were conducted in a single session to reduce inter-test variability. The purpose of this protocol was to quantify the impact of routine clinical interventions on tear osmolarity dynamics, particularly relevant given the device’s sensitivity to environmental variables such as ambient conditions and measurement timing. The examiner performing the tear osmolarity measurements was masked to the participant's group status (normal control versus DED patient) to minimize potential observation bias.”
- Provide a stronger justification for assessing only the right eye, as interocular variability is clinically relevant.
- Thank you for your important comments. We agree with your opinion. We acknowledge that interocular variability is clinically evident, but our methodology was designed to ensure the statistical robustness of the findings. Measurements from both eyes are not statistically independent due to physiological correlation and including both would artificially inflate the sample size and risk introducing paired-data bias in standard analyses. To maintain the requisite independence of observations and simplify the interpretation of our core objective—the effect of SLE/FS—we adhered to the common methodological practice in ophthalmology of analyzing only one eye per participant. In addition, we described the limitation as below. (Discussion 5th Section)
“Third, tear osmolarity was measured only in the right eye; therefore, assessment of inter-eye variability—a recognized diagnostic parameter for DED (interocular difference > 8 mOsm/L)—was not feasible.”
- Results
- Figures and tables are clear; however, consider enlarging labels and adding sample sizes to figures for readability.
- Thank you for your comments. We adjusted the labels and added sample size to Table 2 and Figure 3 as you recommended
Table 2. Changes of tear osmolarity according to slit lamp examination and fluorescein staining
|
|
Group 1 |
Group 2 |
Group 3 |
P value† |
|
Normal controls (mOsm/L) (n=31) |
298.3±11.3 |
299.1±13.3 |
297.0±12.6 |
0.379 |
|
DED patients (mOsm/L) (n=34) |
296.1±11.5 |
296.5±11.0 |
291.2±11.3 |
<0.001 |
Group 1: Baseline; Group 2: After SLE; Group 3: After SLE+FS
SLE: Slit lamp examination; FS; fluorescein stain; DED: dry eye disease
†:Repeated measure ANOVA
- Please highlight the clinical significance of findings, not just statistical significance. For example, how large a change in osmolarity is clinically meaningful?
- Thank you for your comments. We highlighted the clinical significance of finding in the discussion as below. (Discussion 4th section)
“In addition, the clinical significance of our finding is primarily the differential pattern of change between the groups, which reflects impaired tear film homeostasis in DED patients. Specifically, only DED patients experienced a notable drop (4.9 to 5.3 mOsm/L) after SLE+FS. However, when considering established diagnostic criteria, such as the TearLab® interocular difference threshold of >8 mOsm/L, this 5 mOsm/L decrease is unlikely to be a major clinical confounder that would dramatically shift a DED diagnosis or significantly alter patient management.”
Discussion and Conclusion
- The discussion is well developed but could be streamlined for conciseness.
- Thank you for the comments. We corrected the discussion more concisely.
- Emphasize how these results could directly influence clinical workflows (e.g., testing osmolarity before slit lamp examination in DED patients).
- Thank you for the comments. We added the conclusion section as below.
“While these findings support the ideal workflow of measuring osmolarity before SLE+FS to prevent procedure-induced changes, our results also offer a pragmatic approach for busy clinics. Since the mean decrease is relatively small compared to established diagnostic thresholds (e.g., >8 mOsm/L interocular difference), clinicians can cautiously interpret post-SLE+FS readings, as the absolute value still provides meaningful data for tracking DED severity.“
- Limitations are well stated, though the impact of systemic medications/conditions on osmolarity could be more explicitly acknowledged.
- Thank you for the comments. We added the limitations as you recommended.
“Nineth, the impact of systemic medications/conditions on tear osmolarity should be considered.
- Figures/Tables
- Consider simplifying complex plots for broader readership.
- Ensure consistency across figure legends.
- Thank you for the comments. We corrected figure legends more consistently

Reviewer 2 Report
Comments and Suggestions for Authors
In this study, the authors investigated the effects of slit lamp examination (SLE) and fluorescein staining (FS) on tear osmolarity variability. This is an interesting and clinically valuable study. My comments are very minor. Could the authors please consider the following minor issues:
1. Could the authors please provide the width and length of the slit lamp's light used in the study? The manuscript only mentions that it was "utilized at maximal illumination intensity with a 30°angle."
2. As shown in Figure 1, the interval between the second and third steps is 5 minutes. This means the subject underwent two consecutive 20-second slit-lamp examinations within a 5-minute period. However, the OSMOLARITY SYSTEM - USER MANUAL indicates, “Do not collect tear fluid within 15 minutes after a slit lamp examination.”Could this affect the measurement results? Is this time interval considered to be longer than 15 minutes?
3. What are the inclusion and exclusion criteria for the control group? The results showed that the TBUT value in the control group was 6.5±2.8 seconds, which is less than 7 seconds.
4. Additionally, the baseline tear osmolarity value in the control group was 298.3±11.3, which was higher than that in the DED group (296.1±11.5).
Author Response
Reviewer 2
In this study, the authors investigated the effects of slit lamp examination (SLE) and fluorescein staining (FS) on tear osmolarity variability. This is an interesting and clinically valuable study. My comments are very minor. Could the authors please consider the following minor issues:
- Could the authors please provide the width and length of the slit lamp's light used in the study? The manuscript only mentions that it was "utilized at maximal illumination intensity with a 30°angle."
- Thank you for your comment. We apologize for the omission of the specific light dimensions in the Methods section. We used the maximum slit width (14mm) and length (14mm). We added the methods section. (Methods 3rd section)
“The Haag Streit BQ 900 slit lamp (Haag-Streit AG, Bern, Switzerland) was utilized at maximal illumination intensity with a 30° angle between the microscope and illumination system, ensuring optimal visualization of the ocular surface. The slit width and length were maximum (14mm).”
- As shown in Figure 1, the interval between the second and third steps is 5 minutes. This means the subject underwent two consecutive 20-second slit-lamp examinations within a 5-minute period. However, the OSMOLARITY SYSTEM - USER MANUAL indicates, “Do not collect tear fluid within 15 minutes after a slit lamp examination.”Could this affect the measurement results? Is this time interval considered to be longer than 15 minutes?
- Thank you for your comment. We hypothesize that slit lamp examination and fluorescein staining will not induce a statistically significant or clinically meaningful increase in tear osmolarity measurements, and the change in osmolarity and its variability following these procedures will not differ significantly between normal controls and patients with dry eye disease. We acknowledge the short interval is contrary to the manual's conservative 15-minute guideline, but this methodology was deliberately implemented to subject the tear film to a maximal stress test (two consecutive 20-second illuminations every 5 minutes). This rigor allowed us to provide a more robust finding that even under these significantly more challenging conditions, the osmolarity change remained clinically acceptable, supporting the hypothesis.
- What are the inclusion and exclusion criteria for the control group? The results showed that the TBUT value in the control group was 6.5±2.8 seconds, which is less than 7 seconds.
- Thank you for your important comment. New Korean dry eye diagnostic guideline is the combination of ocular symptoms and an unstable tear film (TBUT <=7 s) The control group consisted of participants with an asymptomatic status (Ocular Surface Disease Index (OSDI) score of less than 13) or those with normal tear film function (Tear Film Break-Up Time (TBUT) greater than 7 seconds), or both. This allowed the inclusion of subjects whose mean TBUT was slightly below 7 seconds (6.5±2.8 seconds)
- Additionally, the baseline tear osmolarity value in the control group was 298.3±11.3, which was higher than that in the DED group (296.1±11.5).
- We acknowledge that the baseline tear osmolarity in the control group (298.3±11.3 mOsm/L) was numerically higher than that in the DED group (296.1±11.5 mOsm/L). As is widely recognized, the absolute value of tear osmolarity is highly variable in DED patients, a phenomenon often more pronounced in Asian populations. Given this inherent heterogeneity and variability, especially in early-stage DED (Level 1 or 2 DED patients were included), we believe that clinical interpretation should focus primarily on the change in osmolarity within each individual (intra-subject variability) rather than relying solely on the absolute baseline mean difference between groups. Our core finding—the differential response pattern to SLE+FS—therefore offers a more robust indicator of tear film homeostasis compromise in DED patients.
